# Decentralized TB diagnostic testing with Truenat MTB Plus and MTB-RIF Dx vs. hub-and-spoke GeneXpert MTB/RIF Ultra in Mozambique and Tanzania: a cost and cost-effectiveness analysis

Akash Malhotra[1,2‡]*, Délio Elísio[3‡], Antonio Machiana[4‡], Anange Lwilla[5‡], Jerry Hella[6‡], Neenah Young[2], Celso Khosa[4], Marta Cossa[3], Dinis Nguenha[3], Regino Mgaya[5], Dionisia Balate[4], Mikaela Watson[7], Vinzeigh Leukes[7], Lelisa Fekadu[8], Saima Bashir[9], Adam Penn-Nicholson[7], Morten Ruhwald[7], Leyla Larsson[10], Monisha Sharma[1], Katharina Kranzer[10,11], Claudia M. Denkinger[12,13], David Dowdy[2], on behalf of the T.B. CAPT consortium¶

1 Department of Global Health, University of Washington, United States of America, 2 Department of Epidemiology, Johns Hopkins University Bloomberg School of Public Health, United States of America, 3 Centro de Investigação em Saúde de Manhiça, Mozambique, 4 Instituto Nacional de Saúde, Mozambique, 5 Mbeya Medical Research Center, National Institute for Medical Research, Tanzania, 6 Ifakara Health Institute, Tanzania, 7 FIND, Geneva, Switzerland, 8 Department of Global Health and Population, Harvard TH Chan School of Public Health, United States of America, 9 Manchester Centre for Health Economics, University of Manchester, United Kingdom, 10 Division of Infectious Diseases and Tropical Medicine, University Hospital, LMU Munich, Germany, 11 German Center for Infection Research (DZIF), partner site Munich, Germany, 12 Department of Infectious Disease and Tropical Medicine, Center for Infectious Diseases, Heidelberg University Hospital, Germany, 13 German Center for Infection Research (DZIF), partner site Heidelberg University Hospital, Germany

¶ Membership of the TB CAPT consortia members have been individually listed in the supporting file, S2 Text.
‡ These authors are Co-first authors on this wrok.
* amalhot7@uw.edu

## Abstract

In low-and middle-income countries, missed or delayed tuberculosis (TB) diagnoses contribute to avoidable morbidity, mortality, and transmission. Decentralized testing platforms, such as the Molbio Truenat, may offer solutions by providing accurate point-of-care testing, improving access, and lowering out-of-pocket costs. Despite these advantages, the overall cost and cost-effectiveness of identifying additional TB cases using the Truenat MTB assays remain inadequately explored and understood. We collected economic data from a multicentre randomized controlled trial of TB testing using decentralized Molbio Truenat platform with MTB Plus and MTB-RIF Dx assays (Truenat MTB assays) versus hub-and-spoke Xpert MTB/RIF Ultra (standard of care) in Tanzania and Mozambique (TB-CAPT Core trial). We estimated facility-based diagnostic cost per participant tested and incremental facility-based diagnostic cost per incremental participant initiating TB treatment within seven and sixty days from enrolment. We used the societal perspective and conducted sensitivity analyses to determine key drivers of cost-effectiveness. The facility-based diagnostic cost per

**Data availability statement:** All data can be accessed either in the manuscript text, the supplementary files, or the GitHub repository https://github.com/akash210593/tbcaptcosting

**Funding:** All authors received funding from The European & Developing Countries Clinical Trials Partnership (project number: RIA2017S-2007) for this work. The funders had no role in study design, data collection and analysis, decision to publish, or preparation of the manuscript.

**Competing interests:** The authors have declared that no competing interests exist.

participant initiating treatment within seven days from enrolment in Mozambique was $853(95% uncertainty range: $707, $1072) for hub-and-spoke testing and $690($588, $823) for decentralized testing; in Tanzania costs were $596($485, $746) for hub-and-spoke testing and $592($495, $715) for decentralized testing. At sixty days, costs per treatment initiation were $581($493, $706) for hub-and-spoke vs. $678($576, $811) for decentralized testing in Mozambique, and $391($324, $476) vs. $591($494, $716) in Tanzania. Comparing decentralized to hub-and-spoke testing, the incremental cost per incremental seven-day treatment initiation was $403(-$103, $941) in Mozambique and $580($167, $1638) in Tanzania, and $805(-$10107, $10560) and -$353(-$20299, $20802) for sixty-day treatment initiation, respectively. Utilization (i.e., testing volume) of decentralized equipment was the strongest driver of cost-effectiveness. Decentralized TB testing with Truenat MTB assays is cost-effective relative to hub-and-spoke testing in Mozambique and Tanzania.

## Introduction

Tuberculosis (TB) kills more than 1.3 million people globally each year, with a case fatality rate of 12% [1]. Approximately one-third of all people with TB, have a missed or inaccurate diagnosis [2]. Lack of access to diagnostic testing and participant loss to follow up during the TB care cascade contribute to missed diagnoses, leading to TB morbidity, mortality, and transmission [3]. Diagnostic testing for TB needs to optimize access, turnaround time, and thus time to treatment initiation and cost [4].

"Hub-and-spoke" Xpert MTB/RIF Ultra ("Xpert"; Cepheid, Sunnyvale, CA, USA) testing with specimen transport to centralized laboratories is standard practice for TB diagnosis in many low-and-middle-income countries (LMICs) [5]. While the approach allows for GeneXpert instruments to be placed in central locations, it also results in missed diagnosis and losses to follow up while participants wait for their Xpert results [6,7].

Previous economic evaluations have demonstrated that decentralized testing for TB can be implemented with only a slight increase in per-test cost compared to centralized testing [6]. For example, in peripheral clinics in Uganda, a multicomponent strategy focused on decentralized molecular testing incurred only a 10% increase in per-test cost relative to the standard of care ($20·46 *vs* $18·20) [6]. Modeling studies have similarly suggested that decentralized Xpert testing can be cost-effective in settings characterized by high loss to follow-up, moderate-to-high peripheral testing volumes, and inability to distribute specimen transport costs among other disease entities [8].

More accurate point of care (PoC) testing may increase access and reduce participant out-of-pocket costs [6]. Options for PoC TB testing include Xpert EDGE [9] and the Molbio Truenat MTB Plus and MTB-RIF Dx assays ("Truenat MTB assays"; Molbio Diagnostics, Verna, Goa, India) [10].

The TB-CAPT Core randomized clinical trial assessed the effectiveness of PoC Truenat MTB assays in improving the TB diagnostic pathway in primary health

clinics in Mozambique and Tanzania using a pragmatic design [1]. The primary study outcome was the absolute number and proportion of participants with microbiologically confirmed pulmonary TB starting TB treatment within seven days of enrolment, accounting for potential delays in receiving a diagnostic test due to clinic workflow, patient follow-up, and access barriers. In both Mozambique and Tanzania, a higher proportion of individuals achieved the primary outcome in the Truenat arm vs. the hub-and-spoke standard of care: 7.08% (5.60%, 8.81%) vs. 4.44% (3.16%, 6.05%) in Mozambique, and 7.59% (5.99%, 9.45%) vs. 5.07% (3.86%, 6.51%) in Tanzania [1].

Decentralized testing with Molbio Truenat at peripheral health facilities allows for accurate and rapid, same-visit results, contrasting with the traditional hub-and-spoke model at centralized hospitals or laboratories, thereby improving access and lowering out-of-pocket costs [7]. Despite these advantages, there is a lack of costing studies that evaluate the difference in incremental costs between facility and decentralized testing strategies, which are necessary for decision-makers and budgetary planning.

As part of the TB-CAPT Core study, we collected economic data from the societal perspective to assess facility-based diagnostic costs per participant tested and facility-based diagnostic costs per participant initiating treatment for confirmed TB within seven days of enrolment.

## Methods

### Study design and participants

Cost data were collected within the TB-CAPT Core, a pragmatic, cluster-randomized trial recruiting participants from 26 August 2022, to 16 June 2023, at 29 peripheral health facilities across four sites in Tanzania and Mozambique. The unit of randomization was the clinic. Clinics were eligible for inclusion if they offered TB treatment, did not have a GeneXpert laboratory on-site, and were selected in coordination with the respective National TB Programs. Selection was based on TB notification data from 2018–2019, and clinics were stratified by location and estimated TB testing volume before randomization. Adults presenting with symptoms suggestive of tuberculosis were enrolled across the 29 health facilities, of which 15 were grouped under the Truenat arm where samples were tested using the Molbio Truenat platform and MTB assays (Truenat MTB Plus and MTB-RIF Dx) capable of processing two samples simultaneously on site. The remaining 14 health facilities tested samples according to the hub-and-spoke standard of care strategy, which included Xpert MTB/RIF Ultra testing off-site in all health facilities and additionally in some health facilities sputum smear microscopy on site. The protocol and trial results have been published previously [1].

At intervention clinics, microscopists were trained to use the Truenat platform, and ensure rapid result communication for same-day treatment initiation, with repeat or reflex testing conducted as needed. At control clinics, staff received refresher training on national TB guidelines, with sputum samples processed via smear microscopy on-site or Xpert testing off-site, while operational bottlenecks like sample transport delays and stockouts were addressed through logistical provisions [1].

The study was approved by regulatory and ethical committees in Mozambique (National IRB approval #131/CNBS/22) and Tanzania (National IRB approval #NIMR/HQ/R.8c/Vol.I/2323 and TMDA approval #BD.59/62/46/05). Voluntary written informed consent was sought for enrolment into the study and extraction of participant data.

### Inclusivity in global research

Additional information regarding the ethical, cultural, and scientific considerations specific to inclusivity in global research is included in the supporting file, S1 Checklist.

### Procedures

We estimated the facility-based diagnostic costs of the intervention from the societal (primary) and health system (secondary) perspectives. Across the 29 clinics, we evaluated participant costs from a systematic sample of approximately

every 10th study participant enrolled, including both direct and indirect costs of TB testing. Participant costs were captured through a structured participant cost survey, with interviews directly administered to participants by study staff. To capture health system costs, we conducted project staff interviews, facility assessments, and analysis of trial expense reports.

A bottom-up, ingredients-based, micro-costing approach was used for the cost analysis. Direct participant costs included all medical (consultation fees and any out-of-pocket payment for medicines, X-rays, and diagnostics) and non-medical expenses (travel costs for participants and caregivers, food costs incurred while in-hospital, money spent buying any special foods, etc.). Indirect costs were estimated as the opportunity cost of time spent seeking care (from the time of symptom onset to the time of treatment initiation). The Gross National Income (GNI) per capita at current prices for 2022 were converted to an average hourly wage to estimate economic losses to participants owing to missed work [11].

We estimated the health system costs from a subset (19 out of 29) of intervention and standard of care clinics across the four study sites. Data on TB and HIV testing capacity, sample transportation, testing procedures, testing infrastructure, and related implementation procedures was captured [12]. We classified costs as cartridge, other consumables, communication and monitoring & evaluation (CME), health system staffing, equipment, and warranty and maintenance. Training costs were grouped under CME, while other consumables included sample preparation kits, fuel, personnel protective equipment, and other medical consumables. Capital assets (such as testing equipment) were annuitized and depreciated linearly based on ten expected life-years at a 3% annual discount rate [13]. Costs of equipment were estimated using product catalogues and trial expense reports [14].

Analysis was conducted separately for Mozambique and Tanzania. Facility-based diagnostic costs were captured in the respective local currencies, MZN for Mozambique and TZS for Tanzania, and were converted to 2022 USD using the average exchange rate during the study period (1 USD = 2334 TZS = 63.9 MZN), from August 2022 to June 2023 [15]. The cost analysis used a time horizon corresponding to the trial period, capturing costs incurred during the study.

## Outcomes

The primary effectiveness outcome in the parent TB-CAPT Core trial was the number of participants with microbiologically confirmed pulmonary TB initiating treatment within one week of enrolment. As cost outcomes, we estimated the facility-based diagnostic cost per participant tested, as well as the diagnostic cost per participant starting treatment within seven days and sixty days of enrolment. Our primary cost-effectiveness outcome was the incremental facility-based diagnostic cost per incremental microbiologically positive participant initiating treatment within seven days of enrolment, comparing the Truenat and standard of care arms. The secondary cost-effectiveness outcome was the incremental facility-based societal diagnostic cost per incremental microbiologically positive participant receiving treatment within 60 days of enrolment.

## Analysis

Empirical cost inputs from clinics across the four sites were used to estimate the primary, lower, and upper bound values of each health system cost category. The mean facility-based diagnostic cost per participant tested was calculated from the health system perspective as the total monthly testing cost divided by the number of monthly tests performed at the health facility, assuming 16 tests per month (see supporting file, S1 Text) in the reference case. Participant cost surveys were used to estimate uncertainty ranges of costs from the participant perspective.

We performed a one-way deterministic sensitivity analysis for both Tanzania and Mozambique to evaluate the influence of key assumptions on the incremental facility-based diagnostic cost per incremental microbiologically positive participant initiating treatment within seven days of enrolment. We varied the monthly TB testing frequency from 12 tests per month (one test every other workday) to 125 tests per month (more than five tests per workday). Additionally, we adjusted TB prevalence and modified all input cost categories, each by 25%.

We also conducted a probabilistic sensitivity analysis by sampling each cost and effectiveness value from a corresponding distribution 10,000 times; 95% uncertainty ranges were defined as the 2.5th and 97.5th percentiles of results

across these simulations. In addition to costs, effectiveness values were varied to account for differences in TB prevalence, which influence the number of individuals testing positive and subsequently impact cost-effectiveness estimates. For each parameter, we constructed a beta distribution around the empirically observed point estimate, with the weighted mean serving as the mode of the distribution and range based on the minimum and maximum observed values for each parameter in the trial. We assumed alpha = 4 for all parameters (which corresponds to a 95% confidence interval covering 63% of the width of the full distribution, when the beta distribution was symmetric).

Lastly, we assessed the facility-based diagnostic cost per participant tested at different testing volumes to inform placement and utilization strategies for the dual module Truenat machine. We estimated this cost, from the societal perspective, for both the intervention and standard of care at hypothetical testing volumes assuming a Poisson distribution (ranging from <1% to 100% utilization, see supporting file, S1 Text).

All analyses were performed in Microsoft Excel (version 2311) and R (version 4.3.2).

### Role of the funding source

The funder of the study had no role in the study design, data collection, data analysis, data interpretation, or writing of the report.

### Results

Of 3987 participants in the TB-CAPT Core trial, 388 (9.7%) were enrolled in the participant cost survey. Survey participants were similar to the full study population in most measured characteristics (Table 1). Survey respondents were more likely to be female in Mozambique (Table 2).

### Cost and cost-effectiveness outcomes

From the health system perspective, for the standard of care (hub-and-spoke) arm, the facility-based diagnostic cost per participant tested was $39 (95% credible interval: $36, $43) in Mozambique, and $22 ($19, $24) in Tanzania. The estimated facility-based diagnostic cost per participant tested using on-site Truenat MTB assays was $51 ($46, $56) in Mozambique and $40 ($35, $45) in Tanzania. Fig 1 presents a breakdown of these facility-based diagnostic costs per participant tested.

**Table 1. Participant characteristics for the TB-CAPT Core study and the participant cost survey.**

| | Mozambique | | | | Tanzania | | | |
|---|---|---|---|---|---|---|---|---|
| | Extended participant cost survey* (cost data collected) | | TB CAPT Core full study population | | Extended participant cost survey* (cost data collected) | | TB CAPT Core full study population | |
| | Standard of care | Truenat | Standard of care | Truenat | Standard of care | Truenat | Standard of care | Truenat |
| **Participants enrolled** | 85 | 99 | 855 | 1045 | 110 | 94 | 1125 | 962 |
| **Sex** | | | | | | | | |
| Female | 58 (68%) | 65 (66%) | 489 (57%) | 600 (57%) | 55 (50%) | 44 (47%) | 559 (50%) | 488 (51%) |
| **Age, years** | | | | | | | | |
| Median | 44 | 44 | 43 | 43 | 40 | 40.5 | 42 | 41 |
| 18–30 | 18 (21%) | 20 (20%) | 206 (24%) | 249 (24%) | 23 (21%) | 18 (19%) | 258 (23%) | 210 (22%) |
| 31–40 | 19 (22%) | 21 (21%) | 179 (21%) | 222 (21%) | 33 (30%) | 29 (31%) | 274 (24%) | 254 (26%) |
| 41–50 | 23 (27%) | 19 (19%) | 180 (21%) | 193 (18%) | 17 (15%) | 28 (30%) | 258 (23%) | 240 (25%) |
| >50 | 25 (29%) | 39 (39%) | 290 (34%) | 381 (36%) | 37 (34%) | 19 (20%) | 335 (30%) | 258 (27%) |
| **HIV Status** | | | | | | | | |
| Positive | 43 (51%) | 29 (29%) | 350 (41%) | 288 (28%) | 31 (28%) | 28 (30%) | 358 (32%) | 272 (28%) |
| Negative | 34 (40%) | 42 (42%) | 412 (48%) | 474 (45%) | 64 (58%) | 46 (49%) | 583 (52%) | 509 (53%) |
| Unknown / not tested | 8 (9%) | 28 (28%) | 93 (11%) | 283 (27%) | 15 (14%) | 20 (21%) | 184 (16%) | 181 (19%) |

**Table 2. Cost and cost-effectiveness of decentralized vs. hub-and-spoke testing for tuberculosis in the TB-CAPT CORE trial.**

| | Mozambique | | Tanzania | |
|---|---|---|---|---|
| | Standard of care | Truenat | Standard of care | Truenat |
| Number of participants tested for TB* | 855 | 1045 | 1125 | 962 |
| Societal facility-based diagnostic cost per participant tested**^ | $40 ($37, $44) | $51 ($47, $56) | $31 ($27, $36) | $46 ($41, $52) |
| Total cost | $34,520 | $53,439 | $35,264 | $44,504 |
| Number of 7-day treatment initiations | 38 (27, 52) | 74 (59, 92) | 57 (43, 73) | 73 (58, 91) |
| Cost per 7-day treatment initiation | $853 ($707, 1072) | $703 ($599, $838) | $596 ($485,$746) | $592 ($495, $715) |
| Incremental cost per incremental 7-day treatment initiation | | $422 (-$114, $1019) | | $580 ($167, 1638) |
| Number of 60-day treatment initiations | 57 (44, 73) | 77(61, 95) | 88 (71, 107) | 73 (58, 91) |
| Cost per 60-day treatment initiation | $581 ($493, $706) | $678 ($576, $811) | $391($324, $476) | $591 ($494, $716) |
| Incremental cost per 60-day treatment initiation^^ | | $805 (-$10107, $10560) | | -$353(-$20299, $20802) |

*Number of participants who were offered a test. See supporting file, S1 Text, for details.

**includes both health system and participant level costs

^95% uncertainty ranges are mentioned in parenthesis

^^Scenarios with negative incremental cost per treatment estimates indicate that the Truenat arm is dominated by the standard of care after the 60-day window, meaning that in some simulations, the Truenat arm is more expensive and less effective. This quantity is highly sensitive for the 60-day treatment initiation scenario because when the difference in the number of participants treated is nearly zero, the variability of the estimate is significantly amplified.

As presented in Table 1, from a societal perspective, the facility-based diagnostic cost per participant testing microbiologically positive for TB within seven days of enrolment in the intervention arm was $703 ($599, $838) in Mozambique and $592 ($495, $715) in Tanzania. For the standard of care arm, this facility-based diagnostic cost per participant was $853 ($707, 1072) in Mozambique, and $596 ($485, $746) in Tanzania. The incremental facility-based diagnostic cost per incremental participant initiating treatment within seven days of enrolment was $422 (-$114, $1019) for Mozambique and $580 ($167, $1638) for Tanzania. The incremental facility-based diagnostic cost per incremental 60-day treatment initiation was $805 (-$10107, $10560) for Mozambique and -$353 (-$20299, $20802) for Tanzania.

### Sensitivity analysis

The incremental cost per 7-day treatment initiation ("ICER") was most sensitive to changes in monthly TB testing frequency and equipment and warranty, across both Mozambique and Tanzania. In Mozambique, staffing and monitoring and evaluation (M&E) were also significant cost drivers, while in Tanzania, TB prevalence played a crucial role in affecting the cost-effectiveness. A visual representation of the analysis is presented in the supporting file, Fig A in S1 Text.

### Varying testing utilization

Fig 2 illustrates the societal facility-based diagnostic cost per participant tested at varying testing volumes, a key determinant of cost-effectiveness. Comparing different scenarios of testing volume (from theoretically achievable but not

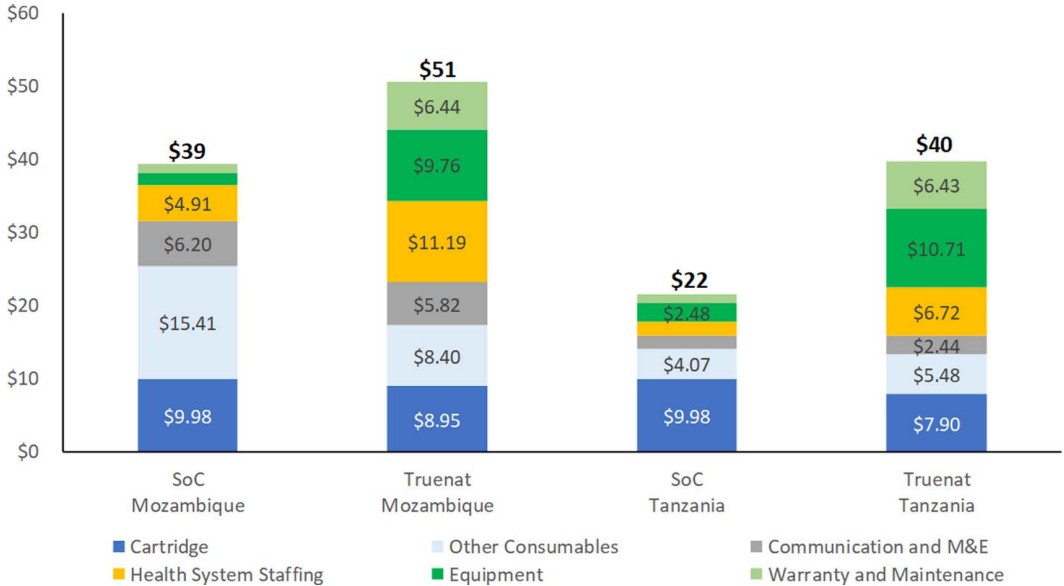

**Fig 1. Health system facility-based cost per participant tested for tuberculosis, comparing point-of-care Truenat MTB assays versus hub-and-spoke Xpert Ultra MTB assays (standard of care, SoC) testing in the TB-CAPT Core Trial.** Each segment within each stacked bar corresponds to a distinct cost category, illustrating its proportionate contribution to the total health system expenses for diagnostic testing in each arm of the trial. Training costs were grouped under communication and monitoring & evaluation (CME), while other consumables included sample preparation kits, fuel, personnel protective equipment, and other medical consumables. All costs are reported in 2022 US dollars. For legibility, costs labels are presented for only those categories where the cost is higher than $2.

practically feasible 16 samples per day to two samples per month), per-test costs stabilized at a volume of ≥3 tests per day, whereas observed testing volumes were 16 per month (0.7 per day) across Tanzania and Mozambique[10]. The supporting file, S1 Text, lists the calculated monthly testing volumes at different utilization scenarios.

On probabilistic sensitivity analysis, the probability of cost-effectiveness at a threshold of $500 per incremental seven-day diagnosis was 0.65 in Mozambique and 0.35 in Tanzania (as presented in a cost-effectiveness acceptability curve, Fig 3). Cost-effectiveness was enhanced under assumptions of higher testing volumes.

## Discussion

The TB CAPT Core study demonstrated that decentralized TB testing via the dual module Molbio Truenat with MTB assays is effective in certain low-resource settings; this analysis complements those results by estimating costs and cost-effectiveness. Across both countries, while the facility-based diagnostic cost per participant tested was higher in the intervention arm ($46 vs $31 in Tanzania; $51 vs $39 in Mozambique), the diagnostic cost per seven-day treatment initiation for confirmed TB was comparable or lower ($592 vs $596 in Tanzania; $703 vs $853 in Mozambique). These results suggest that, in settings where hub-and-spoke molecular testing for TB is currently the standard of care, decentralized testing with Truenat assays is at least similar in cost on a per-rapid-treatment-initiation basis and is associated with better diagnostic outcomes. Decentralized TB testing is most cost-effective at higher testing volumes, in settings with higher prevalence of TB, and in settings with affordable staffing and monitoring and evaluation.

Cost per patient tested provides a useful measure of resource use and efficiency, especially in settings where diagnostic capacity and budget constraints shape decision-making. In decentralized testing models, understanding the cost per test helps guide optimal placement of diagnostic platforms to balance accessibility, affordability, and impact.

At higher testing volumes, our findings of a per test cost of $20-$25 for Molbio Truenat in both countries are consistent with a recent analysis from Uganda, in which the estimated cost of decentralized Xpert testing was $20.46 (in 2022 USD),

A. Mozambique

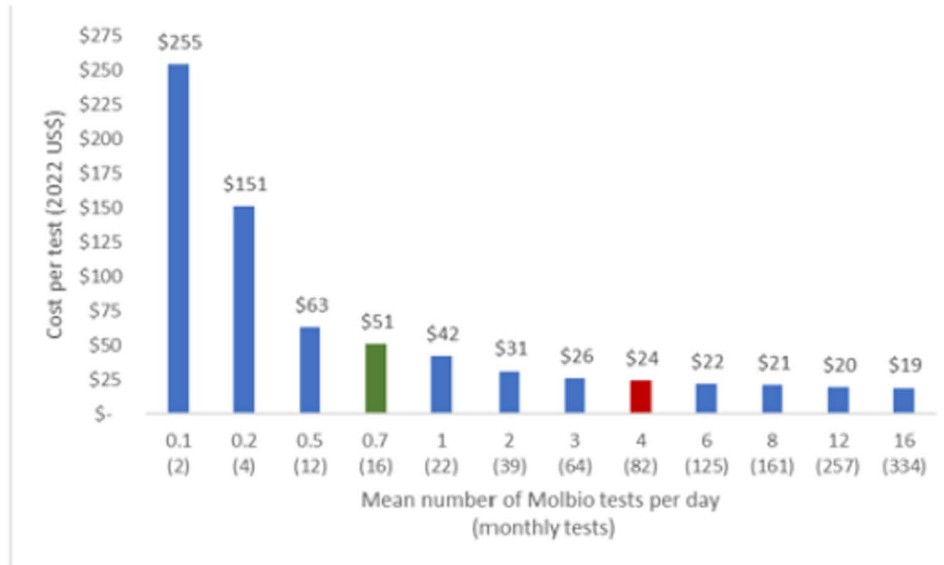

B. Tanzania

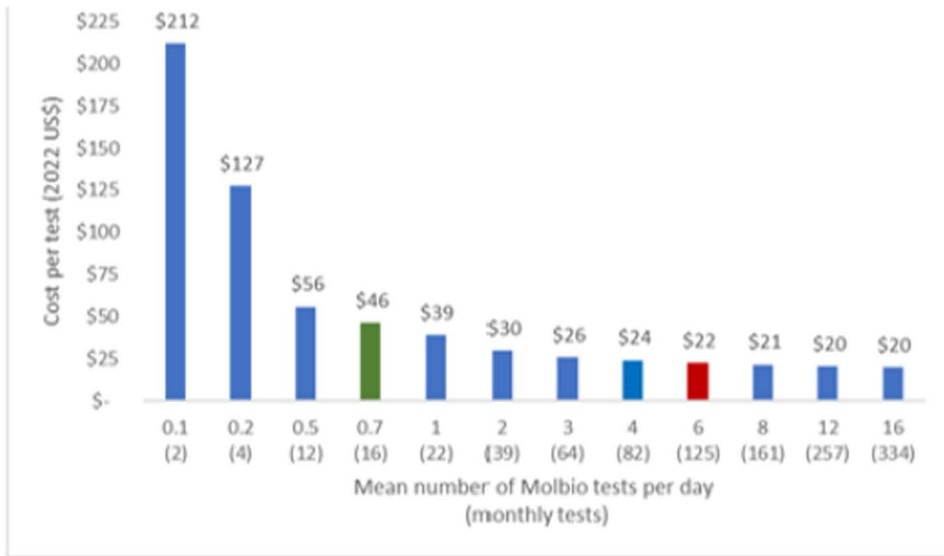

**Fig 2. Unit cost of decentralized testing using Truenat MTB assays for tuberculosis.** Bars show the estimated unit cost (in 2022 US dollars) of decentralized Molbio testing as a function of the mean daily (monthly) number of tests performed (*x*-axis). The green bar indicates the testing volumes observed during the trial. The red bar for each country corresponds to the average number of tests per health facility required to test everyone in the country presenting with symptoms of TB. Maximum capacity for the Molbio MTB/RIF instrument is approximately 16 tests per day. Costs were derived by assuming a Poisson distribution of daily test volumes. Above mean daily volumes of approximately three tests per day, the unit cost remains relatively stable (between $20 and $26 per test).

and of centralized testing was $18.20 [6]. At a willingness-to-pay threshold of $500, the cost-effectiveness of implementing Molbio Truenat improves markedly with increased testing volumes. Specifically, in Mozambique, at a higher testing frequency of three tests per day, the probability of cost-effectiveness reaches 100%, compared to 65% at the currently

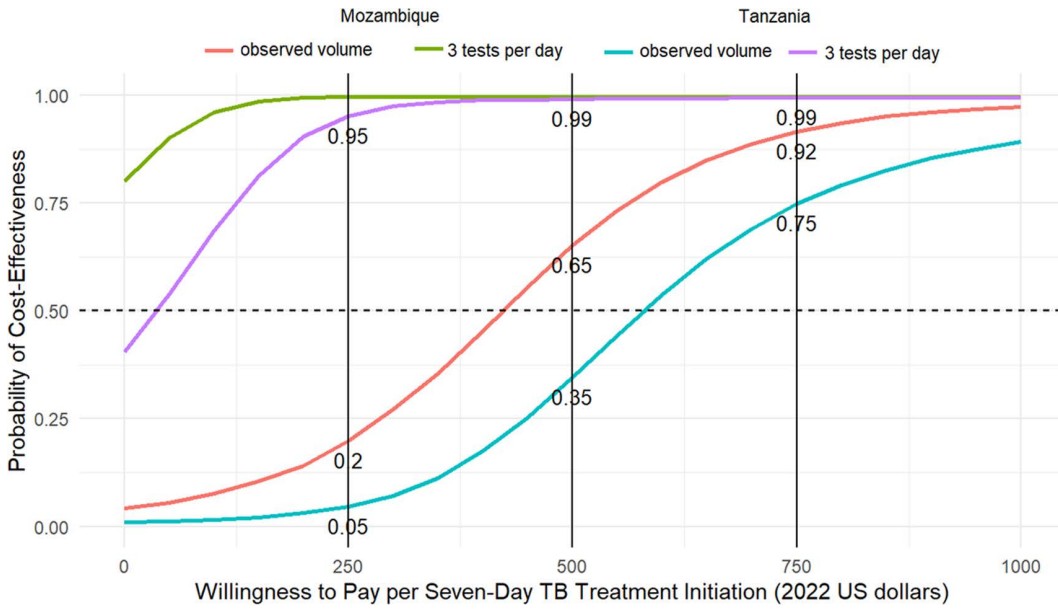

**Fig 3. Cost-effectiveness acceptability curves.** Cost-effectiveness acceptability curves comparing the incremental facility-based diagnostic cost per incremental participant initiating treatment within seven days of enrolment, comparing point-of-care Truenat arm to the hub-and-spoke standard of care. Solid vertical lines represent alternative thresholds for evaluating cost-effectiveness.

observed testing volume of 16 tests per month. In Tanzania, the probability of cost-effectiveness at three tests per day is 99%, higher than the 35% probability observed at the lower testing volume of 16 tests per month.

When considering placement strategies for testing equipment, cost-effectiveness and logistical feasibility may be greatest in areas with higher testing volumes. However, the clinical impact of same-day testing might be most significant in remote clinics with lower testing volumes. As the distance from a clinic to a testing hub increases, as observed with the varying distances to off-site Xpert testing locations in this trial, the cost of transporting samples can become prohibitively expensive. While centralized testing in high-volume areas offers logistical ease and cost-effectiveness, it's essential to recognize the trade-offs between impact and feasibility. In regions with lower testing demand, a decentralized testing model could be crucial in ensuring equitable access to timely care, thereby reducing the burden of multiple trips for testing and results [7].

Among people diagnosed with confirmed tuberculosis in the TB CAPT Core Trial, 82% started tuberculosis treatment on the day of presentation in the intervention compared to 3.3% under the standard of care. Thus, if the primary outcome of the trial were same-day (rather than seven-day) treatment initiation, both effectiveness and cost-effectiveness would be even more favorable for the intervention. We found that when a single module Truenat machine is utilized at higher levels and exceeding 10% of its maximum testing capacity, it demonstrates notable efficiency, leading to a stabilization and plateauing of per-test facility-based diagnostic costs. This suggests that fixed costs such as equipment costs, which are a significant proportion of per-test facility-based diagnostic cost in the Truenat arm have lesser influence on per test costs at scale.

A key strength of this study is that the pragmatic nature of the trial enabling us to estimate costs under programmatic conditions. Another strength is the provision of same-day test results in the Truenat arm, including those testing negative. This allowed healthcare staff to finalize participant interactions immediately, avoiding the need to retrieve participant records later and ensuring that participants did not need to return to the clinic for results, averting out of pocket costs. Owing to the quicker negative diagnosis in the Truenat arm, participants presenting with symptoms specific to TB were more likely to receive a faster clinical diagnosis compared to those in the standard of care.

However, this study also has certain limitations. First, in keeping with the primary effectiveness outcome of the trial, our primary cost-effectiveness outcome was the facility-based diagnostic cost per participant initiating treatment for confirmed TB within a seven-day window. However, the clinical and epidemiological importance of 7-day treatment initiation (as opposed to ever-initiation) has not been fully studied. It is reassuring that the cost per 7-day treatment initiation was similar or lower in the intervention compared to the standard of care, but this estimate may nonetheless be difficult to compare to other cost-effectiveness estimates that use measures such as health utility (Disability adjusted life years, DALYs or quality adjusted life years, QALYs). Second, although implementation costs were collected, they were done so retrospectively. Prospective collection of implementation costs, along with routine time and motion exercised and facility assessments by dedicated costing staff would have strengthened the costing estimates and provided a more complete understanding of costs throughout the implementation process. And lastly, even though this was a pragmatic trial, the conditions of the trial may not fully reflect programmatic reality, particularly in countries with other epidemiological and economic contexts.

The findings of this trial offer valuable insights for policymakers aiming to improve diagnostics infrastructure in resource-limited settings. Future research could consider strategies of placement for limited volumes of Truenat Molbio (or Xpert) instruments, as well as multiplexing of existing testing equipment [16,17]. Other use cases for molecular platforms in low resource settings are Human Immunodeficiency Virus (HIV) and hepatitis B (HBV) viral load testing, sexual transmitted infections, febrile diseases and respiratory viruses specifically influenza and covid-19 [18]. The dependency of cost-effectiveness on test volume reflects all tests performed on a given instrument – not just tests for TB. Thus, if the same instruments could be used to test for other diseases (e.g., HIV, HBV), the cost-effectiveness of the higher testing volume scenarios presented here could be achieved without the need for higher volumes of people with presumptive TB. Recent World Health Organization (WHO) is exploring ways to enhance access to integrated multiplex technologies, such as the Truenat Molbio, and translate them into impactful diagnostic policies [19].

## Conclusion

In conclusion, these results from a pragmatic trial in Tanzania and Mozambique demonstrate that under conditions of high TB prevalence, and high testing demand, decentralized molecular testing for TB using Truenat MTB assays can link more participants to TB treatment within seven days, at a cost per 7-day treatment initiation that is similar to, or lower than, traditional hub-and-spoke testing. Cost-effectiveness could be enhanced further by placing instruments in locations with high TB prevalence and high testing volume – including using the same equipment for diagnosis of multiple diseases. While decentralized testing for TB is likely to be effective and cost-effective, it will require more resources to implement; thus, implementation of this strategy must be accompanied by a careful assessment of existing service volumes, testing infrastructure across disease areas, equity, epidemiology of TB, and budget availability.

## Supporting information

**S1 Checklist. Additional information regarding the ethical, cultural, and scientific considerations specific to inclusivity in global research.**
(DOCX)

**S1 Text. This document provides details on cost data collection, analysis, and classification methods for various input costs, primarily from the health system perspective. Fig A in S1 Text. One-way sensitivity analysis**. Bars present the incremental facility-based diagnostic cost per participant initiating TB treatment within seven days of enrolment comparing on-site Truenat and standard of care (hub-and-spoke) testing under the high (orange) and low (blue) values across each parameter's specified range, holding all other parameters constant. The monthly TB testing frequency was varied from 12 tests per month (every other day) to 125 tests per month (more than five tests per day). All other parameters

were varied by 25%.The vertical line represents the incremental cost-effectiveness ratio when using base-case estimates of all parameters. Only those parameters for which variation resulted in a change of more than $100 in the incremental cost-effectiveness ratio in at least Tanzania or Mozambique are shown for both countries. **Table A in S1 Text**. **Cost Categories.** Outlines the cost categories, items, and calculation methods used in the analysis, with costs allocated per test, per sample, or monthly, and capital costs annualized over their expected useful life. **Table B in S1 Text**. **Ranges for input cost categories.** Lists the point estimate, minimum, and maximum values of each cost category. **Table C in S1 Text**. **Tanzania: Per test cost in 2022 USD, Truenat MTB assays**. Lists the median, lower bound, and upper bound costs per test using the Truenat MTB assays across different utilization scenarios in Tanzania. **Table D in S1 Text**. **Mozambique: Per test cost in 2022 USD, Truenat MTB assays.** Lists the median, lower bound, and upper bound costs per test using the Truenat MTB assays across different utilization scenarios in Mozambique. **Table E in S1 Text.** Estimate of monthly number of tests per facility. Estimates the crude demand for decentralized testing in Mozambique and Tanzania. (DOCX)

**S2 Text.  TB CAPT consortium members.** Lists the members of the TB CAPT consortium and their affiliations. (DOCX)

## Acknowledgments

We acknowledge the healthcare providers and care seekers who participated in this study, the Governments of Tanzania and Mozambique for their support, and the members of the TB CAPT consortium, who have been listed in the supporting file, S2 Text, for their contribution.

## Author contributions

**Conceptualization:** Adam Penn-Nicholson, Morten Ruhwald, Katharina Kranzer, Claudia M. Denkinger, David Dowdy.

**Data curation:** Akash Malhotra, Délio Elísio, Antonio Machiana, Anange Lwilla, Jerry Hella, Marta Cossa, Regino Mgaya, Mikaela Watson, Lelisa Fekadu, David Dowdy.

**Formal analysis:** Akash Malhotra, Neenah Young, Mikaela Watson, Leyla Larsson, Claudia M. Denkinger, David Dowdy.

**Investigation:** Akash Malhotra, Délio Elísio, Antonio Machiana, Anange Lwilla, Jerry Hella, Marta Cossa, Dinis Nguenha, Regino Mgaya, Dionisia Balate, Vinzeigh Leukes.

**Methodology:** Akash Malhotra, Lelisa Fekadu, Saima Bashir, David Dowdy.

**Project administration:** Délio Elísio, Antonio Machiana, Anange Lwilla, Jerry Hella, Celso Khosa, Marta Cossa, Vinzeigh Leukes, Katharina Kranzer, David Dowdy.

**Supervision:** Délio Elísio, Antonio Machiana, Anange Lwilla, Jerry Hella, Celso Khosa, Marta Cossa, Regino Mgaya, Vinzeigh Leukes, Adam Penn-Nicholson, Katharina Kranzer, Claudia M. Denkinger, David Dowdy.

**Validation:** Akash Malhotra.

**Visualization:** Akash Malhotra, David Dowdy.

**Writing – original draft:** Akash Malhotra, David Dowdy.

**Writing – review & editing:** Akash Malhotra, Neenah Young, Dinis Nguenha, Saima Bashir, Adam Penn-Nicholson, Monisha Sharma, Katharina Kranzer, Claudia M. Denkinger, David Dowdy.

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
