## [Decision Letter · Decision Letter 0]

7 Mar 2025

PGPH-D-24-02678

Decentralized TB diagnostic testing with Truenat MTB Plus and MTB-RIF Dx vs. hub-and-spoke GeneXpert MTB/RIF Ultra in Mozambique and Tanzania: a cost and cost-effectiveness analysis

Dear Akash Malhotra

Thank you for submitting your manuscript to PLOS Global Public Health. After careful consideration, we feel that it has merit but does not fully meet PLOS Global Public Health’s publication criteria as it currently stands. Therefore, we invite you to submit a revised version of the manuscript that addresses the points raised during the review process.

We look forward to receiving your revised manuscript.

Kind regards,

Sizulu Moyo, MBCBH, MPH, PhD

Academic Editor

Journal Requirements:

Reviewers' comments:

Reviewer's Responses to Questions

**Comments to the Author**

1. Does this manuscript meet PLOS Global Public Health’s publication criteria ? Is the manuscript technically sound, and do the data support the conclusions? The manuscript must describe methodologically and ethically rigorous research with conclusions that are appropriately drawn based on the data presented.

Reviewer #1: Yes

Reviewer #2: Yes

2. Has the statistical analysis been performed appropriately and rigorously?

Reviewer #1: Yes

Reviewer #2: Yes

3. Have the authors made all data underlying the findings in their manuscript fully available (please refer to the Data Availability Statement at the start of the manuscript PDF file)?

Reviewer #1: Yes

Reviewer #2: Yes

4. Is the manuscript presented in an intelligible fashion and written in standard English?

Reviewer #1: Yes

Reviewer #2: Yes

5. Review Comments to the Author

Reviewer #1: Malhotra and colleagues have examined the costing and cost effectiveness of two TB testing strategies, a decentralised testing approach versus a hub and spoke approach. Both strategies use commercially available NAATs. This was a study nested within the TB CAPT trial, the settings of which were Mozambique and Tanzania. Decentralised TB testing was found to be more cost-effective than a hub and spoke approach. This is an interesting and well-presented manuscript, and the study seems to have been well-done. The findings are a valuable contribution to the TB costing literature, particularly for programs looking at implementing different TB testing strategies. I have some specific comments and questions to strengthen reporting, particularly regarding clarity of outcomes/analysis section.

Comments

Line 54-56: what were the results for 60 days post-enrolment? This is mentioned in the methods, but results aren’t included here so seems a bit incomplete.

Line 74-5: this is a critical rationale for conducting this work, but I am not sure if Reference 6 if the best study to cite here, since this is another costing/modelling study, rather than work that showed that centralised testing resulting in missed diagnoses and patients lost to follow-up. Suggest verifying if this is ideal citation to include and perhaps add another citation to strengthen claim.

Line 93-95: I think it may be helpful to sharpen up terminology in this sentence. I assume that by ‘traditional model’, what is meant is ‘hub and spoke’ – if so, please modify in text. As well, by ‘decentralised’ here, does that mean bringing the tests to people who need testing (e.g., ACF kind of approach), as opposed to passive case finding? Or is it referring to test placement? It seems that simply placing a test in a de-centralised location (e.g., in rural areas) may still require multiple visits for obtaining a TB diagnosis as people may not be able to wait around for an hour+ – so perhaps considering adding a bit of detail to more fully explain what is meant by ‘decentralised’.

Line 106-111: I am not very familiar with TB CAPT (sorry) but wanted to confirm that the unit of randomisation was the clinic/peripheral health facilities? So each clinic included would have at baseline been doing hub-and spoke testing, but would have had the potential to start using TrueNat testing? Perhaps a couple of sentences about the eligibility criteria of the healthcare facilities could be included, to provide a bit more context for readers who aren’t familiar with Mozambique or Tanzania settings.

Line 118-120: Some questions about the participant costs: Can you specify the number of people included as the basis for participant costs? Are you confident that these individuals are representative of the population in the sampling frame? How were the participant costs measured? Survey administered to participant? Survey completed by trial staff? Interview? Something else? Please specify in-text.

Line 130: Some questions about the health facilities selected for the health system costs estimates: how were the 19/29 clinics selected? Was this pre-specified? How many of each arm were included? Do you think these clinics are representative of all studied facilities?

Outcomes: I’m having a bit of a hard time following this section and distinguishing between ideas. There are effectiveness and cost-effectiveness outcomes. Is ‘the facility-based diagnostic cost per participant tested’ and the ‘diagnostic cost per participant starting treatment within seven days of enrolment’ cost or effectiveness outcomes? Something else?

Analysis: Writing is direct and not overly technical. However, after reading through this section a few times, I am still not sure what outcomes/calculations are being discussed in each paragraph:

Line 160: regarding the deterministic sensitivity analysis, ‘evaluate the influence of key assumptions’ on what specific outcome? Mean facility-based diagnostic cost per px tested? Something else? Please specify this in-text.

Line 163-169: For the probabilistic sensitivity analysis, some explanation of assumption(s) being tested would be welcome – in other words, why were these sensitivity analyses done? I think it was to see the impact of varying each cost component, but I don’t quite follow why varying the ‘effectiveness’ value would be helpful (as isn’t effectiveness an outcome? [ie., the number of px with confirmed pulmonary TB starting treatment within 7 days]).

Line 170: how are testing volumes different from testing frequency? Consider indicating in-text

Outcomes + Analysis: perhaps the order of information presented in the Analysis sub-section could follow the order of information in the Outcomes section for improved clarity?

Line 171: I think it may be pertinent to mention earlier than the Truenat machines were dual systems. Perhaps in “study design and participants” section this could be mentioned

Reviewer #2: Peer Review comments: Cost-effectiveness of decentralized TB testing with Truenat vs. GeneXpert

Summary:

The study assessed the cost and cost-effectiveness of identifying additional TB patients using the Molbio Truenat platform with MTB Plus and MTB-RIF Dx assays (Truenat MTB assays), in comparison to the hub-and-spoke Xpert MTB/RIF Ultra system (standard of care). The study highlights the potential for Truenat MTB assays to deliver accurate and rapid, same-visit results for TB diagnosis. The analysis provides valuable insights into the cost-effectiveness of the Truenat MTB assays compared to the Xpert MTB/RIF Ultra system, showing that decentralized TB testing with Truenat MTB assays is cost-effective relative to hub-and-spoke testing Xpert MTB/RIF Ultra system, in Mozambique and Tanzania. However, certain aspects of the study design, particularly the population used, outcome measures, and implementation algorithm, could be clarified further for the reader to follow easily.

Population and Outcomes

It was not very clear to me how the study populations were used to define the outcomes. Specifically, why was the diagnostic outcomes are measured based on individuals enrolled in the study (i.e. line 50;). A clearer explanation of the outcome would be helpful.

• Per patient tested: The report uses this as a primary outcome measure; however, it would be beneficial to discuss the relevance of this metric in assessing cost-effectiveness from a clinical perspective.

• Per patient initiated on treatment within 7-60 days of enrolment: Clarify in the methods why treatment initiation is considered within 7-60 days of enrolment rather than from the point of diagnosis? A clear justification for this choice would strengthen the methodology.

• Implementation Algorithm: It would be useful to have a brief description of the implementation algorithm for the diagnostic tools used in both arms of the comparison. Specifically:

• Population tested: Is the population comprised of passive individuals or TB care-seeking individuals?

• Testing algorithms: What are the specific steps or protocols followed for testing in each arm?

• Clinical performance comparison: Are the Truenat MTB assays and the Xpert MTB/RIF Ultra system comparable in terms of clinical performance, specifically sensitivity and specificity? i.e. if the Truenat assays have lower specificity, it could result in unnecessary treatment being initiated, which should be carefully considered in the cost-effectiveness analysis.

Additional: Do you have results showing patient-level costs (alone) by the different study arm and were there any significant differences between the arms to show that the decentralized TB testing with Truenat MTB reduces patient costs?

Minor:

1. Abbreviations: Spell out some of the abbreviations: DALY, QALY, some diseases.

2. Time Horizon for Costs: I may have missed this, but, the authors can also clarify the time horizon for which the reported estimated costs

6. PLOS authors have the option to publish the peer review history of their article (what does this mean? ). If published, this will include your full peer review and any attached files.

**Do you want your identity to be public for this peer review?** For information about this choice, including consent withdrawal, please see our Privacy Policy .

Reviewer #1: No

Reviewer #2: No

---

## [Editor Report · Decision Letter 1]

13 May 2025

Decentralized TB diagnostic testing with Truenat MTB Plus and MTB-RIF Dx vs. hub-and-spoke GeneXpert MTB/RIF Ultra in Mozambique and Tanzania: a cost and cost-effectiveness analysis

PGPH-D-24-02678R1

Dear Dr Akash Malhotra

We are pleased to inform you that your manuscript 'Decentralized TB diagnostic testing with Truenat MTB Plus and MTB-RIF Dx vs. hub-and-spoke GeneXpert MTB/RIF Ultra in Mozambique and Tanzania: a cost and cost-effectiveness analysis' has been provisionally accepted for publication in PLOS Global Public Health.

Best regards,

Sizulu Moyo, MBCBH, MPH, PhD

Academic Editor

Thank you for addressing the reviewer comments.